# Global record-shattering breadbasket droughts emerge from moderately extreme regional events

Ji Li [1] ✉, Jakob Zscheischler [1,2] & Emanuele Bevacqua [1] ✉

Simultaneous droughts across multiple maize-producing regions can strike record-shattering portions of the global maize agricultural area, threatening global food security as the system is poorly adapted to large shocks. Yet the future probability of such global droughts remains unknown. Here, we close this gap by analyzing surface soil moisture data from large ensemble climate models under future emission scenarios. During 2026-2099, the chance of at least one such event is 52% (32–80%, range across models) under an intermediate emission scenario and 60% (32–100%) under high emissions, about seven to eleven times higher than expected if there were no long-term trends in soil moisture. These elevated probabilities are primarily driven by long-term drying in Brazil, Europe, and the USA. Interestingly, global record-shattering droughts do not emerge from simultaneous regional record-shattering events, but they mostly occur when several regions simultaneously face moderately extreme droughts relative to the new climate. These results demonstrate a high potential for an upcoming global record-shattering drought in crop-producing areas, an under-recognized risk for food security.

Spatially compounding droughts, when multiple major crop-producing regions face soil moisture deficits at the same time, pose a serious threat to global food security[1–5]. Observational studies show that severe droughts affecting individual major agricultural regions can cause substantial crop yield losses and ecosystem productivity declines, and that spatially compounding droughts across breadbasket regions have the potential to reduce global harvests, with impacts on agriculture and food systems[1,6–10]. Although the implications of such events have been widely acknowledged, the likelihood of their most extreme manifestation remains poorly understood. Of particular concern are record-shattering events in which the fraction of the global breadbasket area affected by drought exceeds previous records by a substantial margin (Fig. 1a), marking a step-change beyond past experience and challenging the limits of adaptation capacity[11,12].

Previous studies have evaluated the behavior of record-shattering, or less extreme record-breaking, events for single hazards, such as heat extremes[13], heavy precipitation[14], and marine heatwaves[15].

However, little is known about droughts and compound events under climate change[16], particularly compound droughts spanning multiple breadbasket regions. Accordingly, it also remains unclear how trends in individual crop regions, and associated regional record-shattering droughts, may contribute to global record-shattering events. This knowledge gap limits the ability to anticipate and prepare for rare but potentially high-impact compound droughts in a warming climate.

In this study, we investigate the evolution of the probability of record-shattering droughts under climate change by using surface soil moisture data from multiple Single Model Initial-condition Large Ensembles (SMILEs) that cover the period 1850–2099 and multiple future emission scenarios[17–19]. We focus on droughts across the six major global maize breadbasket regions[2,3,18] (Fig. 1b), which collectively represent most of the world's maize production. The contributions of long-term trends in the mean and variability of surface soil moisture, as well as of trends in individual regions, are disentangled to explain the evolution of global compound droughts. We further investigate

[1]Department of Compound Environmental Risks, Helmholtz Centre for Environmental Research - UFZ, Leipzig, Germany. [2]Department of Hydro Sciences, TUD Dresden University of Technology, Dresden, Germany. ✉e-mail: j.li@ufz.de; emanuele.bevacqua@ufz.de

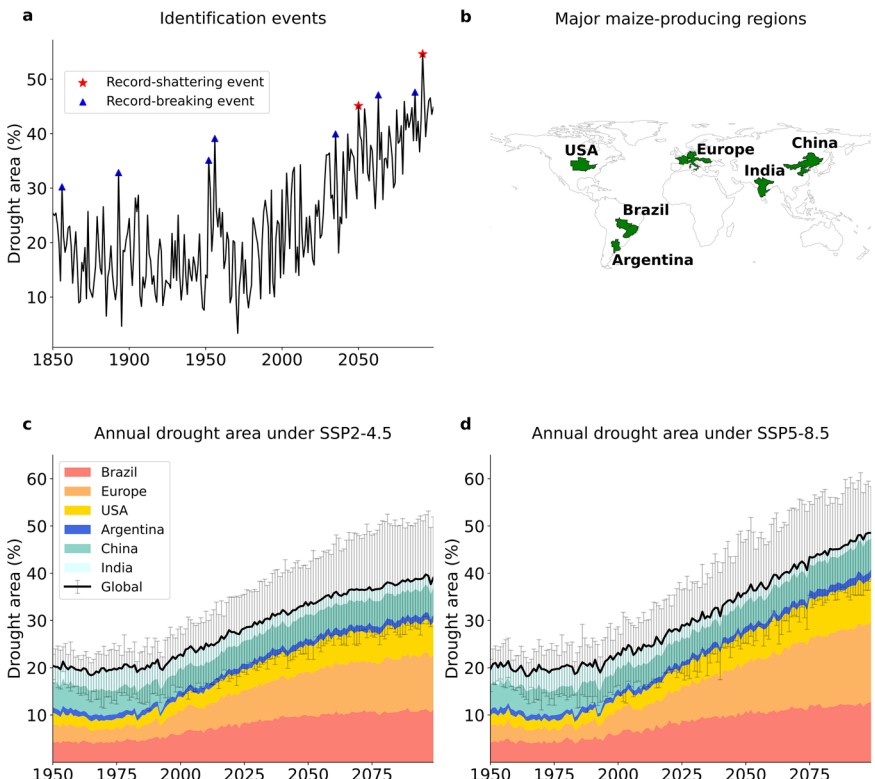

**Fig. 1 | Identification of record-shattering droughts, and the evolution of the regional and global drought areas for the major maize-producing regions.**
**a** Representative time series of annual global drought area (%) derived from a random ensemble member of the model MIROC6. While record-breaking droughts are events that exceed all previous values by any margin (shown in blue), a record-shattering drought occurs when the annual drought area exceeds the previous record by at least 5% (red). **b** The six major maize-producing regions. **c,d** Stacked multi-model mean of the annual drought area for the six regions and globally, defined as the percentage of the global maize area under drought (Methods), under a moderate (SSP2-4.5, panel **c**) and high (SSP5-8.5, panel **d**) emission scenarios. The error bars for the global drought area indicate the inter-model range.

whether global record-shattering drought events tend to occur due to simultaneous record-shattering events across different crop-producing regions.

## Results

### The evolution of the global drought area

The proportion of global maize-growing regions under drought remains relatively stable before 2000 but increases steadily throughout the 21st century (Fig. 1c, d), particularly for the high emission scenario (SSP5-8.5). By the end of the century, nearly half of the global maize growing area is projected to experience drought conditions annually under the higher emission scenario, compared to 38% under SSP2-4.5 (multi-model mean values). Among the six crop regions, the most pronounced increases in regional drought area are projected in Brazil, Europe, and the USA, where the regional drought area is projected to reach approximately 60, 80, and 70% under SSP5-8.5 by the end of this century (multi-model mean values; Supplementary Fig. 1). In contrast, Argentina, China, and India exhibit a slight increase or even a decline in regional drought area under both SSPs. Such little changes in these regions align with increasing precipitation, which can counteract higher temperature-induced evaporation[20,21].

These projections are subject to uncertainties arising from both model differences and internal climate variability[17,18], as reflected by the wide inter-model ranges and ensemble spread (Supplementary Fig. 1). Despite these uncertainties, climate models agree that long-term changes in the mean, rather than interannual variability, of surface soil moisture are the key driver of drought area changes. This is shown by a comparison of the model-simulated changes in drought area with those derived using different detrending experiments, where

the mean or the variability (that is, the standard deviation) of surface soil moisture is held constant (Fig. 2, Methods). At the global scale and across all major regions, under SSP2-4.5, long-term trends in the mean consistently emerge as the main driver of the increase in drought events, while the effect of variability change is comparatively minor. Similar conclusions are obtained under SSP5-8.5 (see Supplementary Fig. 2). This result also implies that uncertainty in long-term trends in mean surface soil moisture is the key contributor to uncertainties in projections of global drought area.

### Probability of global record-shattering droughts

Building on the analysis of drought areas, we assess the temporal evolution of the annual probability of global record-shattering droughts. Under both SSP2-4.5 and SSP5-8.5, the annual probability of a global record-shattering drought—an annual global drought area that breaks previous records by at least 5%—rises from the start of the 21st century, generally peaks around mid-century, and then levels off or declines slightly toward the end of the century (Fig. 3). The timing of the peak differs between SSP scenarios, with the peak occurring later under SSP5-8.5, reflecting more sustained drying trends in the latter half of the century under the higher emission pathway. While uncertainty in the estimated record-shattering probabilities exists due to model differences and internal climate variability (Supplementary Fig. 3), the annual probability of global record-shattering droughts during the century is substantially higher than expected under stationary surface soil moisture (gray line, "Detrend both mean and standard deviation"; see Methods). Overall, this highlights that new record-shattering global droughts are likely to occur in the future, events that would not have been possible without human-induced

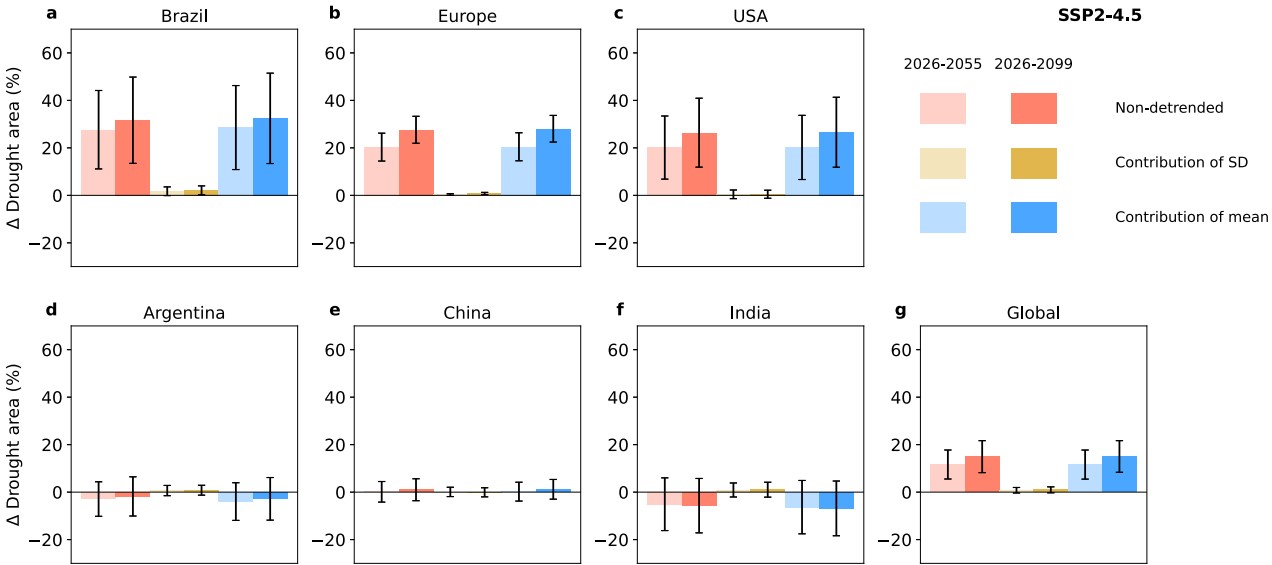

**Fig. 2 | Projected changes in annual drought area for the six major maize-producing regions and the global scale. a–f** The projected changes for individual regions under SSP2-4.5 for two future time periods (2026-2055, light and 2026-2099, dark) relative to the baseline (1850-1899) based on original data (non-detrended, red), and the contribution from changes in surface soil moisture standard deviation (SD, yellow) and mean (blue) (see Methods). Bars represent the multi-model mean, and error bars indicate the inter-model range based on the ensemble mean of each model. **g** As in the individual regions, but aggregated to the global scale.

climate change (compare red and gray lines, Fig. 3 and Supplementary Fig. 3).

The temporal evolution of record-shattering annual probability mirrors the trend in drought area (Fig. 1), where a rapid increase in drought area in the early 21st century leads to a surge of record-shattering events. In contrast, as the increase in drought area slows and stabilizes later in the century, record-shattering events become less frequent, resulting in a reduced record-shattering annual probability. These results align with previous work that demonstrated a similar relationship between the rate of change in climate variables and associated record-shattering probabilities[11,16]. Accordingly, detrended simulations highlight that the general evolution of global record-shattering drought annual probability is primarily driven by long-term trends in the mean of surface soil moisture rather than by changes in variability (Fig. 3).

As adaptation decisions are usually made on a multi-decadal horizon, we assessed the probability of experiencing at least one global record-shattering drought during the next three decades (period 2026-2055 in Fig. 4a and Supplementary Fig. 4a, red bars). The probability exceeds 28% according to the multi-model mean both under SSP2-4.5 and SSP5-8.5, which is substantially higher than expected if there were no trends in surface soil moisture (gray bars in Fig. 4a and Supplementary 4a, about 3% on average). When considering the whole future part of the century, that is 2026–2099, the probability of experiencing at least one global record-shattering drought reaches 52% (32–80%, range across climate models) under SSP2-4.5 and 60% (32–100%) under SSP5-8.5, about seven to eleven times higher than the expected under stationary surface soil moisture. These probabilities underscore a high chance of experiencing at least one global record-shattering drought by the end of the century (Fig. 4b and Supplementary Fig. 4b).

## Regional contributions to global record-shattering droughts

We disentangled the contribution of surface soil moisture trends in each of the six major maize-producing regions to the probability of having at least one global record-shattering drought (Fig. 4). Retaining surface soil moisture trends in Brazil, Europe, or the USA alone (while detrending all other regions) yields a probability well above zero—meaning that surface soil moisture trends in these regions are the key drivers of the increase in the probability. In contrast, leaving Argentina, China, or India undetrended has little effect, meaning that surface soil moisture trends in these regions contribute minimally. These regional contributions are consistent with the corresponding trends in regional drought areas, with Brazil, Europe, and the USA also showing the most pronounced increases (Fig. 1). Given the relatively flat or even declining drought area trends in Argentina, China, or India throughout the 21st century (see Supplementary Fig. 1), their contribution may even slightly dampen the overall risk. Note that the late-century decline in record-shattering probability is due to a slowdown or stabilization of drying trends instead of a saturation of the drought area, as most models project that drought areas remain well below 100% by the end of this century (Supplementary Fig. 1).

Surface soil moisture trends in Europe, the USA, and Brazil not only increase drought areas in these regions, but also raise the probability of regional record-shattering droughts during the century (Supplementary Fig. 7). As a result, global record-shattering droughts may be expected to emerge due to underlying simultaneous regional record-shattering droughts. However, results under SSP2-4.5 reveal that approximately 73% of global record-shattering droughts occur without any underlying regional record-shattering event (Fig. 4c). This fraction increases to about 82% under SSP5-8.5 (Supplementary Fig. 4c), and decreases to approximately 42% when defining local droughts based on the more restrictive 5th percentile threshold rather than the 20th percentile (Supplementary Fig. 5c). When a regional record-shattering drought is involved, it is typically confined to a single region—mainly Europe, the USA, or Brazil. Consequently, global record-shattering droughts are rarely driven by two or more regions simultaneously under regional record-shattering droughts (2.1% under SSP2-4.5; 1.3% under SSP5-8.5; and 6.3% under SSP2-4.5 with the 5th percentile definition).

By contrast, most of the global record-shattering events are driven by simultaneous moderately extreme droughts in multiple regions—specifically, droughts that fall among the 20% largest observed in the 30 years preceding the global record-shattering event (Fig. 4d). This

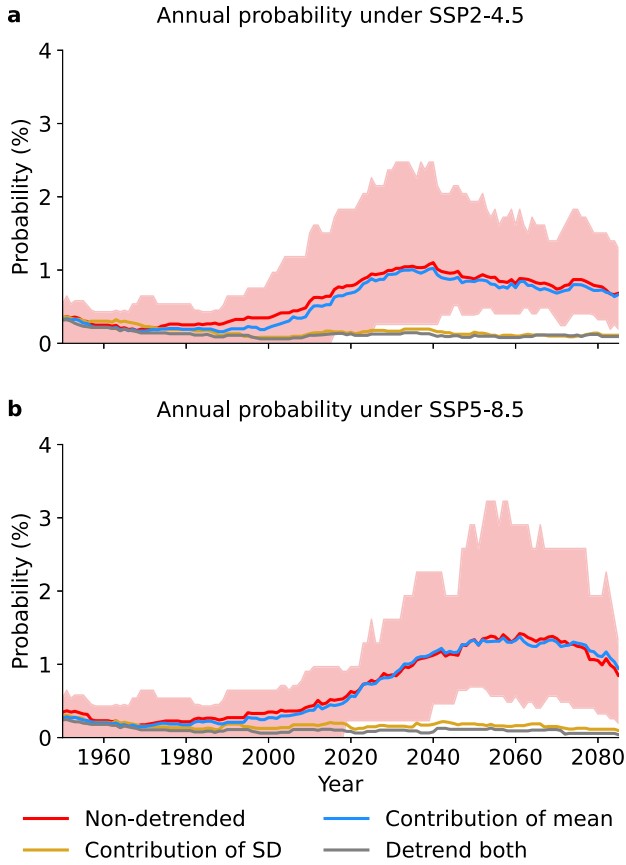

**Fig. 3 | Annual probability of global record-shattering drought. a** Probability during 1950–2085 under SSP2-4.5 based on the original data (non-detrended), where the solid red line shows the multi-model mean and red shading shows the inter-model range. Other lines show the contribution to the probabilities from changes in surface soil moisture's standard deviation (SD, yellow), mean (blue), and both mean and standard deviation (gray) (Methods). **b** As in (**a**), but for SSP5-8.5.

highlights how a combination of moderately extreme conditions in individual regions can lead to global record-shattering conditions, potentially with global-scale ramifications for the agricultural sector. Furthermore, although regional record-shattering events rarely occur in the exact same year as global record-shattering events, many regional record-shattering events cluster within 15 years of global record-shattering events without aligning precisely in time (Fig. 4e). These findings are robust against alternative drought definitions, different choices of moderate drought thresholds, and different emission pathways. Specifically, analyses using a more restrictive 5th percentile drought threshold, varying the definition of moderate regional drought, and considering the SSP5-8.5 scenario yield qualitatively similar behavior (Supplementary Fig. 4–6), underscoring that, regardless of drought severity or future emissions, global record-shattering droughts are generally expected to emerge from the aggregation of moderately extreme regional droughts rather than from perfectly synchronized regional record-shattering events.

## Discussion

By targeting droughts that exceed historical spatial extents across the world's major crop-producing regions by large margins, this study provides key insights into a class of extreme events that remain undetected by conventional drought metrics. Global record-shattering droughts pose severe threats not only to food production systems but also to economic stability, global trade, and social resilience. Despite

uncertainties in climate projections, the results indicate a marked increase in the probability of global record-shattering droughts, both under an intermediate emissions pathway that closely resembles current policy trends[22] and a high emission scenario, with probabilities well above expectations under stationary conditions. While due to data availability our analysis focuses on six SMILEs, which is relatively large for analyses of record-shattering probabilities, we note that a broader sampling of models could further expand the range of uncertainty. The probability of at least one global record-shattering event by the end of the century—that is, during 2026–2099—is 52% (32–80%, range across climate models) under SSP2-4.5 and 60% (32–100%) under SSP5-8.5. Such probabilities are even higher when considering record-breaking events, reaching approximately 95% on average across the two SSPs (see Supplementary Fig. 8). Furthermore, drought spatial extent and drought intensity are tightly coupled (Supplementary Fig. 9), meaning that our considered record-shattering spatially extended droughts are expected to coincide with substantial negative anomalies in soil moisture. We find that the projected rise in record-shattering drought probability is predominantly driven by long-term declines in mean surface soil moisture in key breadbasket regions, with changes in surface soil moisture variability playing a minor role. Our results align with recent evidence that anthropogenic warming has already increased global drought severity by roughly 40% worldwide[23], including climate change contributions to a recent multi-breadbasket soybean failure[24], and underscore growing risks to global food production as maize-producing regions have a high potential to face droughts far beyond historical experience.

Beyond quantifying the probabilities of record-shattering droughts, this study offers mechanistic insights into how such record-shattering events are likely to emerge. Although regional record-shattering events are projected to rise in individual maize-producing regions, most global record-shattering droughts do not coincide with simultaneous regional record-shattering events. Instead, they emerge from simultaneous moderately extreme, spatially extended droughts across multiple regions. This pattern might be preferable to a world in which global record-shattering events are driven by simultaneous regional record-shattering droughts. However, it also highlights serious challenges, as while moderately extreme regional events may be manageable individually, their simultaneous occurrence poses systemic global risks that could overwhelm trade, storage, and emergency response capacities[4,25,26]. Furthermore, we note that while these regional droughts will be only moderately extreme relative to the ongoing future climatology, they may still qualify as severe extremes relative to present-day climatic conditions, reinforcing the risks posed by their concurrence. Our decomposition experiments demonstrate that trends in Brazil, Europe, and the USA contribute most strongly to the emergence of global record-shattering droughts, whereas agricultural areas in Argentina, China, and India exert limited effects. These regional asymmetries suggest that regions exhibiting limited drying could serve as strategic buffers within the interconnected global food system. In particular, strengthening the production and trade capacity of these regions may help build resilience against systemic drought risk under climate change.

The prospect of a widespread global record-shattering drought across major maize breadbaskets underscores the importance of integrating hazard assessments with agricultural impact assessments. Such assessments should carefully account for climate model uncertainties, particularly given that each of the considered SMILEs agrees with at least one of three observational and reanalysis soil moisture datasets in terms of drought area variability and trends over the last decades—the two key drivers of record-shattering probabilities (Supplementary Figs. 12,13; see Methods). Moreover, surface soil moisture deficits are a key stressor for crops, but yield losses can arise from the interplay of multiple factors[27], including irrigation availability[28], nutrient limitations[29], and $CO_2$ fertilization effects[30,31]. Heat stress during

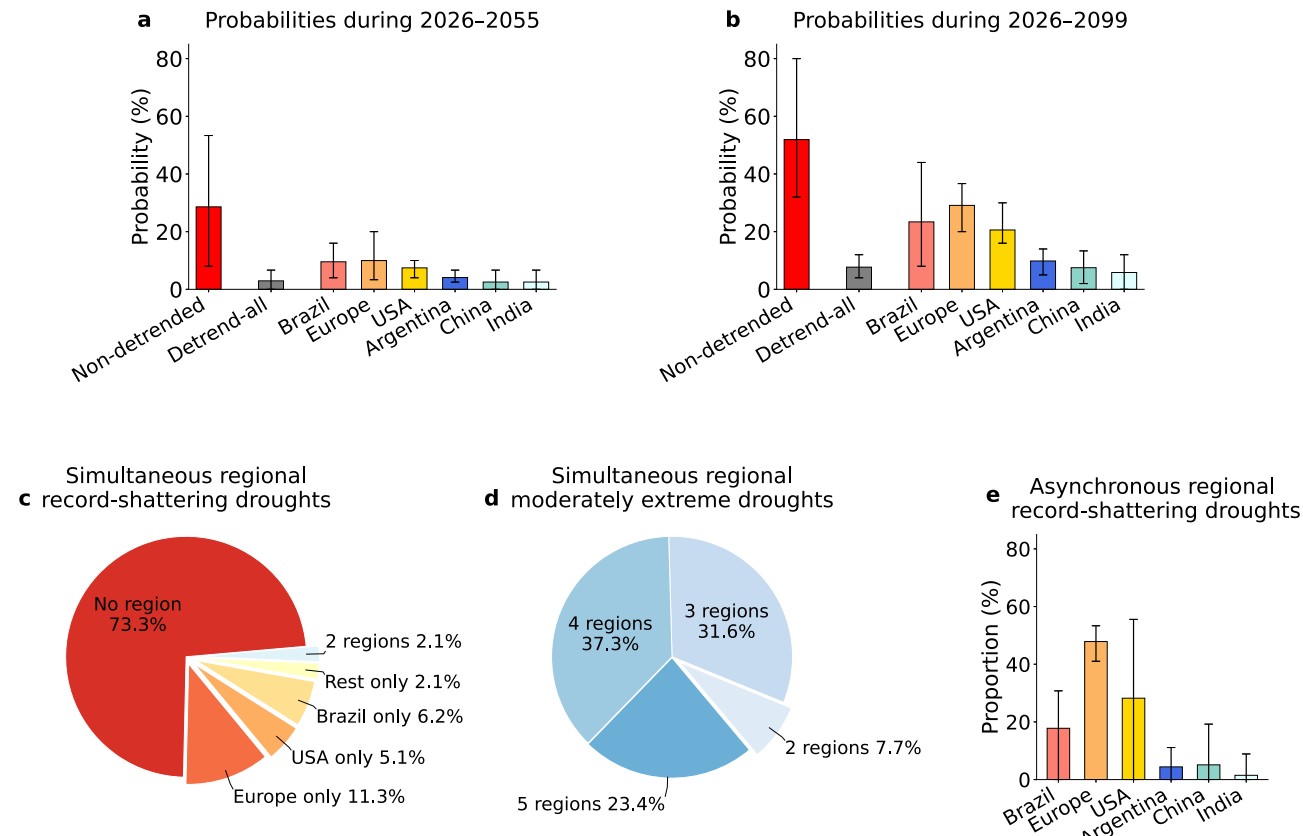

**Fig. 4 | Contributions of regional surface soil moisture trends and timing of regional droughts relative to global events under SSP2-4.5. a** Multi-model mean probability of the occurrence of at least one global record-shattering drought during 2026-2055 (red bar) and the same probability under stationary surface soil moisture in all crop regions (gray bar). The other bars represent the contribution of soil-moisture trends in each crop region (obtained by detrending surface soil moisture in all other regions; Methods). Error bars represent inter-model range. **b** As panel **a**, but for the probability during 2026–2099. **c** Multi-model mean of the proportion of global record-shattering droughts during 2000-2099 that coincide with no regions, only one region ('Rest only' means the sum of the remaining cases of only one region), or multiple regions simultaneously under regional record-shattering droughts—categories with 0% proportions are not shown. **d** As (**c**), but for cases where global record-shattering droughts coincide with regional moderately extreme droughts (see Methods). **e** Multi-model mean of the proportion of global record-shattering droughts accompanied by a record-shattering drought in a given region (x-axis) within 15 years before or after the global event year; error bars represent inter-model range.

sensitive growth stages can further exacerbate crop losses[32], making it particularly concerning that soil moisture droughts could coincide with record-shattering heatwaves[24,26]—projected to become more frequent in the near future[11]. Overall, our results stress the importance of employing well-calibrated crop models to systematically assess the consequences of a plausible upcoming global record-shattering drought. Such assessments are essential for anticipating systemic threats to food security and for guiding strategies that enhance resilience across interconnected global food systems.

## Methods
### Data
We derived surface soil moisture (10 cm depth, name of the variable *mrsos*) from SMILEs of the Coupled Model Intercomparison Project Phase 6 (CMIP6)[33], by combining historical simulations (1850–2014) with future projections (2015–2099) under the Shared Socioeconomic Pathways SSP2-4.5 and SSP5-8.5[34]—the two SSPs represent intermediate-emissions and a high-emissions pathways, respectively. Each SMILE comprises many ensemble members, which are independent realizations of the same climate model under identical forcing in a given SSP; the members differ only due to internal variability. SMILEs enables the estimation of probabilities for record-shattering compound events[11,18]. Previous studies show that robust probability estimates for compound events and record-shattering events require a

sufficiently large number of realizations[11,18,35]; accordingly, we selected models with at least 10 ensemble members, which we consider sufficient for reliable probability estimation. The employed models are: GISS-E2-1-G (25 members under SSP2-4.5, and 14 under SSP5-8.5), MIROC-ES2L (30, 10), MPI-ESM1-2-LR (30, 30), ACCESS-ESM1-5 (40, 40), CanESM5 (50, 50), MIROC6 (50, 50).

Before further analyses, surface soil moisture is conservatively regridded to a uniform 2.5° × 2.5° grid to facilitate inter-model comparison. To account for differences in crop-growing seasons across hemispheres, we use annual surface soil moisture, calculated as the average surface soil moisture over each calendar year. Surface soil moisture provides a physically grounded indicator of agricultural drought, as it directly governs plant water stress and land-atmosphere fluxes and is therefore closely linked to drought impacts in a changing climate[36,37].

To evaluate the realism of simulated drought area, we use three widely used observational and reanalysis soil moisture datasets: ERA5 surface soil moisture from the first soil layer (0–7 cm; reanalysis), GLEAM root-zone soil moisture (0–10 cm; satellite-driven retrievals combined with land-surface modeling), and MERRA-2 surface soil moisture (0–5 cm; reanalysis)[38–40]. These datasets differ in how they were derived (methodology, forcing, and model structure), together providing a representative range of observational uncertainty. Their

effective soil layer depths are broadly consistent with the surface soil moisture variable used in the SMILEs (mrsos; 0–10 cm).

The global maize breadbasket in this study covers six major production regions, namely the USA, Brazil, Argentina, Europe, China, and India, which together—in 2012—accounted for 74% of maize production within breadbasket countries and 56% of global maize production[2]. The specific provinces and states within each region are listed in Gaupp et al.[2].

### Global and regional record-shattering droughts, and their probability

For each SMILE, we first identified grid cell-based droughts as events in which annual surface soil moisture falls below the local 20th percentile of the surface soil moisture distribution derived from the pooled ensemble members during the reference period (1850–1899). This percentile-based definition corresponds to moderate drought conditions as commonly defined in drought classification frameworks and provides a consistent threshold for identifying spatially extensive droughts while retaining a sufficiently large sample size for robust statistical analyses[41,42]. Then, to track the evolution of spatially compounding droughts, we define the *global annual drought area (%)* as the fraction of global maize breadbasket area under drought in a given year. For each region, we also define the *regional annual drought area (%)* as the regional area under drought; to capture both local and global perspectives, this was calculated as a percentage relative to both the total regional area or relative to the global maize area.

In our analysis, we focus on the drought area. In Supplementary Fig. 9, to characterize drought intensity associated with spatially compounding droughts, we define the global mean drought intensity as the area-weighted mean of standardized surface soil moisture anomalies across all grid cells classified as under drought in a given year. Surface soil moisture anomalies were computed relative to the preindustrial reference period (1850-1899) and standardized by dividing by the corresponding standard deviation ($\sigma$) at each grid cell.

A spatially compounding, global record-shattering drought event is defined as a global annual drought area that exceeds the previous maximum within a given ensemble member by an area equal to or larger than 5% of the total breadbasket area (Fig. 1a), representing a substantial absolute exceedance of the previous record. Sensitivity analyses illustrate that the choice of exceedance margin affects the characterization of global record-shattering droughts: a higher margin (7.5%) drastically reduces the events across models (Supplementary Fig. 10a–f), whereas a lower margin (2.5%) yields probabilities comparable to ordinary record-breaking droughts (Supplementary Fig. 10g–l). However, the probability patterns remain consistent: in all cases, soil moisture trends in Brazil, Europe, and the USA emerge as the dominant drivers of the global record-shattering drought (Supplementary Fig. 11). The chosen 5% margin threshold balances these cases, being sufficiently large to capture truly extreme record-shattering events while still allowing a clear separation from record-breaking ones. Regional record-shattering droughts are defined accordingly, when a regional annual drought area exceeds its previous record by at least 5%. To estimate the annual probability of record-shattering events (Fig. 3 and Supplementary Fig. 7), a 31-year moving window (±15 years) is used. Within each window, the number of record-shattering events is divided by the total number of years across all ensemble members (that is, 31 times the number of the considered ensemble members), yielding a time-varying probability of record-shattering events across all ensemble realizations within each 31-year window.

For both regional and global events, to estimate the probability of experiencing at least one record-shattering drought within a given period (2026-2055 or 2026-2099), for each SMILE, we calculated the fraction of ensemble members that experience at least one record-shattering event during that period (Fig. 4). In the same way, we

estimated the probability of having at least one global record-breaking drought, where the exceedance margin of all previous events is set to zero (Supplementary Fig. 8).

### The role of trends in the mean and variability of surface soil moisture to droughts

To evaluate the respective roles of long-term trends in the mean and variability of surface soil moisture in driving the time-evolution of the annual global drought areas and annual regional drought area (Fig. 2), and global record-shattering event probabilities (Fig. 3), we applied a detrending procedure[43] to the annual surface soil moisture time series of each grid cell before further analysis.

First, for each grid cell, we pooled all ensembles of a given model and then computed the mean and standard deviation (SD) of annual surface soil moisture within 31-year moving windows. These are referred to as the pooled mean and standard deviation and were used to implement three distinct detrending experiments, as described below.

1. Detrend mean: for each ensemble member, grid cell, and year, the pooled 31-year moving-window mean (centered on the considered year) is subtracted from the original time series to remove long-term changes in the mean. The resulting anomalies are then shifted by adding the mean of the preindustrial reference period (1850–1899). This produces a detrended time series that preserves the original standard deviation while maintaining a stationary mean equal to that of the preindustrial baseline.

2. Detrend standard deviation (variability): for each ensemble member, grid cell, and year, the pooled 31-year moving-window mean (centered on the considered year) is subtracted from the original time series. The resulting anomalies are then normalized by dividing each year's anomaly by the corresponding pooled 31-year moving-window standard deviation, and subsequently rescaled by multiplying by the pooled standard deviation of the preindustrial reference period. Finally, the pooled moving window mean is added back to retain the original mean evolution. This procedure removes changes in the standard deviation while maintaining the original long-term mean trend.

3. Detrend both mean and standard deviation: for each ensemble member and grid cell, this combined approach removes the long-term mean trend by subtracting the pooled 31-year moving-window mean from each year's value, and then normalizes the resulting anomalies by dividing each year's anomaly by the corresponding pooled 31-year moving-window standard deviation. Finally, the pooled preindustrial mean is added to the whole time series, resulting in a time series that maintains both the mean and standard deviation of the preindustrial reference period.

To quantify changes in global and regional drought areas, we computed the difference between mean annual drought areas over a future period (2026–2055 or 2026–2099) and the corresponding mean over the preindustrial reference period (1850–1899) (Fig. 2). These changes were derived from the model's original (non-detrended) data and data from each of the three detrending experiments (we computed for each model the mean across ensemble members; based on these, we show the multi-model mean and the inter-model range). For Fig. 3, annual probabilities of global record-shattering droughts were computed using the same set of data. Results from experiments 1-3 allow for isolating the individual contributions of changes in variability (experiment 'Detrend mean'), long-term mean trends (experiment 'Detrend standard deviation'), and their combined effects (experiment 'Detrend both mean and standard deviation') to the evolution of both regional and global drought areas, as well as global record-shattering probabilities.

## Decomposition of regional contributions

In Fig. 4a, b and Supplementary Fig. 4a, b, we assessed the influence of trends in surface soil moisture in individual regions on the occurrence of at least one global record-shattering drought within a given 30-year period by employing a region-wise detrending approach. Specifically, to quantify the influence of the trends in a given region, we left its surface soil moisture time series unchanged while detrending both the mean and standard deviation for all other regions, following the approach described in the previous section. Then, the influence of the trend in that region is defined as the difference in global record-shattering drought probabilities between this leave-one-region-undetrended experiment and the detrend-all case. Based on the modified time series, we recomputed the probability that at least one global record-shattering drought occurs during the future time periods (2026–2055) and (2026–2099), using the multi-model mean probability and representing the inter-model range based on the ensemble mean of each model.

In Fig. 4c–e and Supplementary Fig. 4c–e, we inspect whether global record-shattering droughts occurring between 2000 and 2099—using data during the whole century increases the sample size relative to analyses starting in 2026, enabling a more robust assessment. Specifically, we examine whether these global record-shattering events (1) coincide—that is, occur in the same year—with (simultaneous) regional record-shattering droughts, (2) coincide with (simultaneous) moderately extreme regional droughts, and (3) are associated with regional record-shattering droughts occurring within a $\pm 15$-year window centered on each global event.

For (1), we calculate the fraction of global record-shattering drought events that coincide with: no regions under record-shattering droughts; one single region only under record-shattering drought; two regions only simultaneously under record-shattering droughts; and so on, until six regions simultaneously under record-shattering droughts. Note that these categories are mutually exclusive, therefore, their associated fractions add up to 100%. Furthermore, we disaggregated the category "only one single region under record-shattering drought" into four sub-categories: Europe only, USA only, Brazil only, and Rest only (where "Rest" comprises all other study regions).

For (2), we proceed as in case (1), but instead of looking for regional record-shattering events, we focus on regional moderately extreme droughts. For a given region, a year $Y$ of a given ensemble member is defined as under a regional moderately extreme drought (relative to the ongoing climatology) when the regional drought area exceeds the 80th percentile of the distribution derived from year $Y - 30$ to year $Y$ of the analyzed ensemble member. Note that as for the rest of the analysis, the definition of moderately extreme drought relates to the drought's spatial extent and whether a grid point is under drought or not depends on the fixed surface soil moisture percentile-based threshold from the preindustrial baseline.

The case (3) serves to ensure that the finding that global record-shattering droughts often emerge from moderately extreme regional droughts is not simply due to the absence of regional record-shattering events around the year of occurrence of each global event. To this end, we quantified the fraction of global record-shattering droughts for which a regional record-shattering drought occurs within 15 years before or after the global event. Since the global events span from 2000 to 2099, this window-based analysis is restricted to the period 2000-2085 to ensure that the full $\pm 15$-year window lies within the study period.

## Evaluation against observational and reanalysis soil moisture datasets

In general, the evaluation of the probability of record-shattering events is challenged by the limited temporal coverage of observations and the rarity of the events. For this reason, here, we compared SMILEs simulated drought area variability (standard deviation) and trends, which are the two key drivers of record-shattering probabilities, against observations and reanalysis. Furthermore, because drought is defined relative to a preindustrial reference period (1850–1899), direct observation by taking into account the baseline preindustrial period is not possible. We therefore assess the realism of simulated drought area by comparing SMILEs with observational and reanalysis datasets over the overlapping period of data availability, that is, 1980-present. Accordingly, we select 1980–2009 as a reference period to define grid-point droughts and compute annual drought area for 1980–2024 for each dataset and each SMILE model.

This approach enables a consistent evaluation of the interannual variability and long-term trends in drought area at both global and regional scales (six major breadbasket regions). Linear trends in annual drought area (%) are estimated for each ensemble member and summarized at the model level for comparison with observational estimates. The uncertainty ranges shown for the observational and reanalysis datasets in Supplementary Fig. 12 correspond to the standard deviation ($\sigma$) of the estimated drought-area trends. Interannual variability is quantified as the standard deviation ($\sigma$) of annual drought area over the evaluation period (Supplementary Fig. 13).

Overall, the comparison indicates that each SMILE model agrees with at least one observational dataset in every region and globally, with simulations generally falling within the spread of observational uncertainty (Supplementary Figs. 12,13). These results highlight the importance of carefully considering the uncertainties in our projected probabilities of record-shattering droughts, even more as a broader sampling of models could further expand the range of projections[44].

## Data availability

CMIP6 data can be retrieved at https://esgf-data.dkrz.de/projects/esgf-dkrz/. ERA5 reanalysis data were obtained from the Copernicus Climate Data Store (https://cds.climate.copernicus.eu/datasets). GLEAM soil moisture data are available at https://www.gleam.eu/. MERRA-2 reanalysis data can be accessed via the NASA Global Modeling and Assimilation Office at https://gmao.gsfc.nasa.gov/gmao-products/merra-2/. Global breadbaskets can be derived from Gaupp et al.[2].

## Code availability

All analyses were performed using Climate Data Operators (CDO)[45], Python and Bash scripts, following standard methods and techniques, as detailed in the Methods.

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

## Acknowledgements

This project received funding from the Deutsche Forschungsgemeinschaft (DFG, German Research Foundation) via the Emmy Noether Program (Grant ID 524780515). J.Z. acknowledges funding from the European Union's Horizon 2020 research and innovation program (Grant Agreement No. 101003469) and the Helmholtz Initiative and Networking Fund (Young Investigator Group COMPOUNDX, Grant Agreement VH-NG-1537). The authors acknowledge the World Climate Research Program, which, through its Working Group on Coupled Modeling, coordinated and promoted CMIP6, and the authors thank the climate modeling groups for producing and making available their model output, the Earth System Grid Federation (ESGF) for archiving the data and providing access, and the multiple funding agencies who support CMIP and the ESGF. Analyses were carried out on the High-Performance Computing (HPC) Cluster EVE, a joint effort of both the Helmholtz Center for Environmental Research -UFZ and the

German Center for Integrative Biodiversity Research (iDiv) Halle-Jena-Leipzig.

## Author contributions

The study was conceived by E.B., with J.L. J.L. carried out the data analyses and implementations. E.B. supervised the research and provided guidance throughout. J.L. and E.B. wrote the paper. All authors (E.B., J.L., and J.Z.) discussed the results and reviewed the manuscript.

## Funding

## Competing interests

The authors declare no competing interests.
