## [Transparent Peer Review file · Nature Communications]

Global record-shattering breadbasket droughts emerge from moderately extreme regional events

Corresponding Author: Dr Ji Li

Version 0:

Reviewer comments:

Reviewer #1

(Remarks to the Author)

The paper evaluates future occurrence of simultaneous droughts affecting the main maize producing regions of the world. In this it addresses a significant and under-explored issue of climate hazard affecting resilience of the global food supply system. Analyses presented in the paper are methodologically sound, structure is appropriate, language is very good, figures are of good quality, conclusions are justified by the analyses.

I recommend minor revisions.

My main concerns are that analyses are implemented around several subjective choices about data sources and definitions of considered events. While those choices might not affect the overall conclusions of the paper, they are not necessarily well explained and justified in the manuscript.

Firstly, it is not clear what is the motivation for using SMILES instead of the entire CMIP6 ensemble. SMILES are intended to provide information about the level of internal variability or initial condition uncertainty, contrasting it with model uncertainty expressed by the entire ensemble. I do not see how that information is used in the paper. Conclusions are not in any way linked to the process of internal variability and rather target model uncertainty which is presented in the figures in the form of ensemble range. Considering that the SMILES ensemble has only 6 models while the entire CMIP6 ensemble has over 30, the analyses as presented in the paper certainly underestimate model uncertainty. I don't see any step of the analyses that could not be applied to projections from models having only 1 ensemble member. Even the analyses of probability of experiencing at least one record-shattering drought within a given period could be carried out on a single ensemble member, albeit with binary outcome - i.e. that probability would be either 0 or 100%. If there are no compelling reasons for using SMILES instead of a full CMIP6 ensemble (which I might have missed), then redoing the paper with the full ensemble is required as using SMILES only is not really defensible in the context of the objectives of the paper.

Secondly - the definitions of drought levels, apart from the record-breaking one, seem rather arbitrary. For record-shattering - why 5%? Why not 4 or 6? For moderate drought - why 80th percentile? Is there a particular motivation for these values? if not - then please clearly articulate that this is just an arbitrary choice.

More generally - why are record-shattering events significant, compared to record-breaking? Or even compared to "standard" "stationary" definitions such as SPI/SPEI classes? I know that authors allude to the importance of record-exceeding events by bringing up the temporal aspect of adaptive capacity, but perhaps a paragraph presenting some evidence of this emerging from literature would be needed. Additionally, it would be good to illustrate some actual examples of observed events of the type analysed in the paper. Did such droughts occur in the past? What were their global impacts?

Minor remarks:

- is the record-shattering drought threshold of 5% expressed in absolute % or relative %, i.e. calculated with respect to total area of a breadbasket? or area under previous record drought?
- why some analyses are carried out for the period after 2026? and some for the period after 2000?
- what is the motivation for using +/- 15 year window in the analyses of global vs. regional drought? Why not just +15 years? A regional drought occurring after a global one has likely a stronger impact than the other way around, no?
- global drought is calculated as the area-weighted percentage of regional droughts. Shouldn't it be more sound to calculate

it as a production-weighted percentage?

Reviewer #2

(Remarks to the Author)

Review for NCOMMS-25-78075

Title: Emerging global record-shattering breadbasket droughts from regional moderately extreme events

Authors: Ji Li, Jakob Zscheischler, and Emanuele Bevacqua.

Reviewer Recommendation: Major Revisions

General Summary:

This manuscript, submitted to Nature Communications, presents research to assess potential changes to the likelihood of global drought events specific to regions that produce a large portion of the global food supply. The authors primarily use simulations from an ensemble of global climate models to determine that the chance of a “record-shattering” global drought event is from 52-60% by the end of the century. The authors point out the lack of research on simultaneous drought events in different parts of the globe and put forward their methods as an approach to address this knowledge gap.

While a necessary gap to fill, there are some important details lacking in the analysis as well as some missing justification and missing clarifications needed to enable replication of the work by others and potentially reproduction for this or other kinds of extreme events. The edits required necessitate major revisions. My major comments are below, followed by specific comments at given lines.

Major Comments

1. The study relies entirely on model simulated trends in drought area. While the study briefly touches on the questions of uncertainty and the interannual variability component, there is no discussion, nor even a mention, of this ability of these GCMs to capture an observed record-breaking or “record-shattering” event. Indeed, the wide range from the GCM ensembles in the historical periods shown in Extended Data Figure 1 for example suggest that some of these GCMs likely do not capture the historical frequency of such events, or (more to the definition the authors use) the historical drought area. Simply performing an analysis of trends in drought area from model data and providing no insight to the validity of the model simulations for the target metrics gives no guidance for the validity of the conclusions made by the authors. There should be some evaluation of how well this ensemble of GCMs replicates observed trends in drought area and the frequency of record breaking droughts. Also, since the six GCMs used may replicate relevant physical processes better in some regions than others, it is quite relevant to see how well each GCM captures the drought area in each of the breadbasket regions.
2. Related to the above, there is no justification given for why the six specific GCMs are used. Perhaps this is related to the ability of the GCMs to capture critical processes related to soil moisture and drought, but the reader is left to wonder since there is no justification given. In a similar vein, the authors describe their drought classification as being when the annual surface soil moisture falls below the 1850-1899 local 20th percentile. An interesting choice for how to classify a drought, but why use the 20th percentile? Why not the 10th or 5th percentile? Similarly, why is a “record-shattering” drought when an event exceeds previous drought area values by at least 5%? Why not 2.5% or 10% or another value? Also, why the 10cm soil moisture over the soil moisture from other depths? The appropriate justification for each of these should be included in this manuscript. In addition, my questions, among others also present useful opportunities for future work related to the sensitivity of the results. For example, if the threshold of a drought was the local 5th percentile over the 20th percentile, would the results presented here be significantly different? Such options for future work and analysis could be discussed in the results section.
3. The key conclusions depend on how the authors define the probability of a record-shattering event. First, the authors define a record-shattering event as being related to drought area, rather than the drought intensity. This is interesting, and I do wonder why this choice to focus on drought area for this definition without regard to intensity. Under this definition, a record-shattering drought could be one that covers a significantly large area, but where the classification for drought in all of the region effected is barely met. Is it really record-shattering if the area effected is larger, but the intensity does not change (or changes little)? There was no explanation given for the focus on area over intensity, or better still, area alongside intensity. Please add an explanation for that to the methods. Second, the definition of the annual probability of a record-shattering event is perhaps incorrect (perhaps in this case because the awkward phrasing of the definition has left me unclear). The authors state that the probability each year, with the 31-year moving window, is “The number of record-shattering events is divided by the total number of annual drought area values across all ensemble members (that is, 31 times the number of the considered ensemble members), yielding a time-varying probability of record-shattering events.” This sentence is quite confusing, but the interpretation I had reading this is that the probability is defined as the number of record-shattering events divided by the number of drought events. If this is correct, then the actual value defined is the probability of a given drought event being a record-shattering event, rather than a probability of a record-shattering drought occurring at all. This definition needs to be clarified for readers, and analysis of the probability should reflect this definition.

Specific Comments

Lines 55-57: “This is shown by a comparison of the actual changes...” – The figures are not showing the actual observed changes, but the modeled changes from one random model. Please adjust the text, but also describe which model is used

for Figure 1a.

Lines 92-94: "Retaining surface soil moisture...increase in probability." – Why detrend some but not all?

Version 1:

Reviewer comments:

Reviewer #1

(Remarks to the Author)

I have carefully examined authors' responses to my comments, and am happy with how they are addressed. I have no further comments, and recommend the manuscript to be accepted as is.

Reviewer #2

(Remarks to the Author)

This is the second time I have reviewed this manuscript. The authors have taken into account the comments of the reviewers on the original version of the manuscript. This revised manuscript is greatly improved over the first version. I have no further suggested revisions for this manuscript and I look forward to seeing the revised manuscript in print.

Response to Reviewer1

We thank the referee for the careful and constructive review. The comments and suggestions were valuable and have improved the manuscript. We have revised the paper accordingly. Reviewer comments are shown in *italics*, responses in regular text, and changes to the manuscript in **bold**.

Reviewer:

“Firstly, it is not clear what is the motivation for using SMILES instead of the entire CMIP6 ensemble. SMILES are intended to provide information about the level of internal variability or initial condition uncertainty, contrasting it with model uncertainty expressed by the entire ensemble. I do not see how that information is used in the paper. Conclusions are not in any way linked to the process of internal variability and rather target model uncertainty which is presented in the figures in the form of ensemble range. Considering that the SMILES ensemble has only 6 models while the entire CMIP6 ensemble has over 30, the analyses as presented in the paper certainly underestimate model uncertainty. I don’t see any step of the analyses that could not be applied to projections from models having only 1 ensemble member. Even the analyses of probability of experiencing at least one record-shattering drought within a given period could be carried out on a single ensemble member, albeit with binary outcome - i.e. that probability would be either 0 or 100%. If there are no compelling reasons for using SMILES instead of a full CMIP6 ensemble (which I might have missed), then redoing the paper with the full ensemble is required as using SMILES only is not really defensible in the context of the objectives of the paper.”

Response:

We thank the reviewer for raising this point. In the following, we clarify why it is key to use SMILEs (a SMILE is a model that provides multiple realisations). Our study aims to estimate the probability of record-shattering compound droughts, which are extremely rare. To obtain meaningful probability estimates, previous studies on probabilities of record-shattering events have consistently employed SMILEs (Liu et al., 2025; de Vries et al., 2024; Fischer et al., 2021). Fischer et al. (2021), by focusing on heatwaves at the daily scale, showed that a substantial fraction of the model spread actually reflects internal variability rather than true model differences and that large ensembles are therefore required for this type of analysis. Such relevance of using SMILEs is even more important for our analysis of annual-scale events, where the aspect of the sample size becomes prominent. Furthermore, SMILEs are even more important for record-shattering compound droughts, given that previous studies demonstrated that a sufficiently large number of realizations is important to robustly study compound events (Bevacqua et al., 2023).

While in principle the steps of the analyses could be applied to projections from models having only 1 ensemble member, as indicated by the referee, with only one realisation per model, each model would yield a probability of either 0% or 100% for the occurrence of a record-shattering event. Such binary outcome based on a single member directly reflects the non robust probability estimate and the large probability uncertainties discussed above. In fact, as shown by our SMILE-based analysis, the robustly estimated probability of each model falls in between the 0-100% range. This implies that multiple members are required to robustly estimate the probability for each model (and thus distinguish the estimates of the different models robustly). In particular, regarding model uncertainty, based on such model-dependent binary outcomes for each model, the range across models in the estimated probabilities would not provide robust information on uncertainties, as such range could have three possible outcomes: 0-0%, 0-100%, or 100-100%. Given that the real probability of each model falls in between this range, such uncertainty range would not be robust. For example, if by chance single members are selected from each model that lead to estimating a 100-100% or 0-0% range, this could be highly misleading as giving a false sense of certainty. While a 0-100% would most likely represent an overestimation of the actual uncertainties. Finally, we also note that the use of SMILEs allows us to provide insights into contribution to uncertainties from internal variability versus those associated with model differences for the evolution of the drought area (Extended Data Figure 1).

Overall, for the reasons above, it is key to employ available model output with multiple ensemble members. Within the CMIP6 models, six models provide more than ten ensemble members, and based on our analysis needs, we consider this ensemble size the minimum required to derive reliable probability estimates. This requirement necessarily limits the number of available models, however we also note that the resulting set of six SMILEs is comparatively large in the context of previous studies that assessed record-shattering event probabilities (Liu et al., 2025; de Vries et al., 2024; Fischer et al., 2021), which typically employed four to five models. While these six models span a range of model structures and physical formulations, providing model diversity while satisfying the ensemble-size requirements of our analysis, we acknowledge that they cannot capture the full spectrum of model uncertainty. That is, despite our effort of using a relatively large number of SMILEs compared to previous studies, we agree that uncertainties in projected probabilities may be even larger than estimated here.

We have therefore added a statement in the Discussion where we discuss model uncertainties, to emphasize that: **“While due to data availability our analysis focuses on six SMILEs, which is relatively large for analyses of record-shattering probabilities, we note that a broader sampling of models could further expand the range of uncertainty (Bevacqua et al., 2026).”**

We also added a sentence in the Methods section stating that **“Previous studies show that robust probability estimates for compound events and record-shattering events require a sufficiently large number of realizations (Bevacqua et al., 2023; Fischer et al., 2023; Fischer et al., 2021); accordingly, we selected models with at least 10 ensemble members, which we consider sufficient for reliable probability estimation”**.

Reviewer:

“Secondly - the definitions of drought levels, apart from the record-breaking one, seem rather arbitrary. For record-shattering - why 5%? Why not 4 or 6? For moderate drought - why 80th percentile? Is there a particular motivation for these values? if not - then please clearly articulate that this is just an arbitrary choice.”

Response:

1. Why we define the record-shattering margin as 5%.

We thank the reviewer for raising this relevant point regarding the choice of the record-shattering margin. As suggested by the referee, we clarify the choice.

Furthermore, to comprehensively address the comment, we also assessed the sensitivity of our results to this choice by performing additional analyses using thresholds around 5%. When the threshold is increased to 7.5%, the probability of global record-shattering droughts becomes very small across all models (typically below 0.5% over 1950–2100; see Response Figure 1). This reflects the fact that a 7.5 % exceedance lies well beyond the range of interannual variability in drought area simulated by the models, hence a 7.5% margin does not provide a useful metric for distinguishing exceptionally large drought area anomalies. Note that to ensure that our probability estimates are statistically robust for such rare events, we also applied a bootstrap approach (Methods), which confirms that probabilities remain extremely small for a 7.5% threshold.

In contrast, setting the margin to 2.5% produces the opposite behavior (Response Figure 2): the estimated probabilities rise substantially in all models. A 2.5% exceedance falls well within the typical range of interannual fluctuations and leads to results that tend to be more similar to those obtained for ordinary record-breaking events. Taken together, we chose a 5% margin because it is large enough to capture record-shattering events, while still clearly separating them from more modest record-breaking ones.

To further assess robustness, we repeated the regional contribution analysis shown in paper Figure 4 a, b using the alternative 7.5% and 2.5% margins (Response Figure 3). Despite the substantial differences in event frequency under these two sensitivity settings (Response Figures 1 and 2), the multi-model probability patterns remain consistent: in all cases, soil moisture trends in Brazil, Europe, and the USA emerge as the dominant drivers of the global record-shattering drought. These results confirm that the key attribution findings presented in the paper are stable and do not depend on the exact threshold used to define a record-shattering event.

To fully address this comment, we added the following sentence to the Methods section: **“Sensitivity analyses illustrate that the choice of exceedance margin affects the characterization of record-shattering droughts: a higher margin (7.5%) drastically reduces the events across models (Extended Data Fig. 10a–f), whereas a lower margin (2.5%) yields probabilities comparable to ordinary record-breaking droughts (Extended Data Fig. 10g–l). However, the probability patterns remain consistent: in all cases, soil moisture trends in Brazil, Europe, and the USA emerge as the dominant drivers of the global**

record-shattering drought (Extended Data Figure 11). The chosen 5% margin balances these cases, allowing for capturing record-shattering events while providing a clear separation from record-breaking ones.”

Response Figure 1. **Annual probability of global record-shattering droughts with record-shattering events defined based on a margin of 7.5%.** a-f, Probability under SSP2-4.5 based on different models (stated in the title of the panels). The red line shows the probability based on the original (non-detrended) data (consistent with other record-shattering probabilities in the paper, this is based on pooled ensemble members of each model). For each model, shading shows the centered 95% confidence intervals derived via bootstrap (1000 resamples of the ensemble members of the model), representing uncertainty from internal variability.

SSP2-4.5

Response Figure 2. Same as Response Figure 1, but for a 2.5% margin.

Response Figure 3. **Contributions of regional surface soil moisture trends to global record-shattering droughts based on different margins.** **a**, Multi-model mean probability of the occurrence of at least one global record-shattering drought with a margin of 7.5% during 2026-2055 (red bar) and the same probability under stationary surface soil moisture in all crop regions (gray bar). The other bars represent the contribution of soil-moisture trends in each crop region (Methods). Error bars represent inter-model range. **b**, As panel **a**, but for the probability during 2026-2099. **c,d**, As **a,b** but for the margin of 2.5%.

2. Why we define the moderately extreme droughts based on the 80th percentile.

Regarding the definition of moderately extreme regional droughts in Figure 4d, we used the 80th percentile and avoided a higher threshold to maintain the selection of moderately extreme regional events (note that very extreme, record-shattering regional events, are instead considered in Figure 4c). However, we agree that the exact choice of the threshold is somewhat arbitrary. To fully address the comment, we tested the robustness of the choice.

We repeated the analysis using both the 50th percentile threshold (identifying much milder droughts) and the 90th percentile threshold (identifying more severe droughts). Response Figure 4 illustrates the sensitivity of the detected occurrence of simultaneous regional droughts (under global record-shattering events) to the threshold. As expected, the lower threshold (50th percentile) classifies moderate drought more frequently and therefore yields a larger fraction of

global record-shattering events with simultaneous regional droughts, whereas the higher threshold (90th percentile) results in fewer coincident regions. Despite these differences, the main conclusions remain robust: global record-shattering droughts are predominantly associated with simultaneous moderately extreme droughts across multiple regions. Thus, we maintain the 80th percentile represents a reasonable middle-ground threshold for characterising moderately extreme droughts.

Response Figure 4. **Sensitivity of the simultaneous regional drought patterns to the definition of moderately extreme droughts, under SSP2-4.5.** **a.** Multi-model mean proportion of global record-shattering droughts during 2000–2099 that coincide with regional droughts defined using the 50th percentile threshold. **b.** As panel **a**, but using the 90th percentile threshold.

Furthermore, we have now explicitly incorporated this clarification into the main text by jointly summarizing the sensitivity tests to alternative drought definitions, different thresholds for moderate regional droughts, and emission scenarios in a single robustness statement: “**These findings are robust against alternative drought definitions, different thresholds for moderate regional droughts, and different emission pathways. Specifically, analyses using a more restrictive 5th percentile drought threshold, varying the definition of moderate regional drought, and considering the SSP5–8.5 scenario yield qualitatively similar behaviour (Extended Data Figures 4, 5 and 6), underscoring that, regardless of drought severity or future emissions, global record-shattering droughts are generally expected to emerge from the aggregation of moderately extreme regional droughts rather than from perfectly synchronized regional record-shattering events.**”

Reviewer:

“More generally - why are record-shattering events significant, compared to record-breaking? Or even compared to “standard” “stationary” definitions such as SPI/SPEI classes? I know that authors allude to the importance of record-exceeding events by bringing up the temporal aspect of adaptive capacity, but perhaps a paragraph presenting some evidence of this emerging from literature would be needed. Additionally, it would be good to illustrate some actual examples of observed events of the type analysed in the paper. Did such droughts occur in the past? What were their global impacts?”

Response:

We agree that it is a good idea to expand on this matter. Record-shattering events are relevant because they represent excursions far beyond previously experienced variability, rather than marginal updates of historical records. This distinction is critical for impacts and risk, as adaptation, infrastructure design, and preparedness are typically calibrated to past extremes and may fail when confronted with unprecedented magnitudes. Recent studies emphasised that record-shattering extremes can trigger disproportionate impacts, precisely because they fall outside societal, institutional, and ecological experience (Fischer et al., 2021; Kelder et al., 2025). Accordingly, literature has started to emerge on the topic of record-shattering events, which can be identified by dynamically exploring time series (that is, as in Figure 1a) (de Vries et al., 2024; Fischer et al., 2021; Thompson et al., 2022). Note that, in principle, SPI or SPEI could be used as a basis to define record-shattering events, however they are typically used in a “standard stationary” fashion (as indicated by the referee), preventing a focus on record-shattering events.

It is also a good idea to expand on observed examples and impacts. While direct observational examples of record-shattering spatially compounding droughts have not yet been explicitly documented, evidence exists for severe impacts associated with extreme droughts occurring within individual major agricultural regions. Such events have been shown to cause substantial crop yield losses, as well as ecosystem impacts, and societal consequences across Africa, North America, Europe, and East Asia (Kiros, 1991; Liu et al., 2018; Li et al., 2019; Buras et al., 2020). Notably, Anderson et al. (2019) show that large scale climate anomalies in 1983 were associated with concurrent maize yield reductions across multiple major maize-producing regions. In addition, a growing body of literature highlights the risk of spatially compounding droughts across breadbasket regions, with the potential for synchronous impacts on agriculture and food systems (Singh et al., 2018; Singh et al., 2021; Qi et al., 2022). These studies motivate our focus on record-shattering, spatially compounding drought events.

To address this comment, we added a sentence at the beginning of the main text stating that **“Observational studies show that severe droughts affecting individual major agricultural regions can cause substantial crop yield losses and ecosystem productivity declines, and**

that spatially compounding droughts across breadbasket regions have the potential to reduce global harvests, with impacts on agriculture and food systems (Singh, D. et al., 2018; Anderson et al., 2019; Singh, J. et al., 2021; Qi, W. et al., 2022; Singh, J. et al., 2022; Li et al., 2026). Although the implications of such spatially compounding droughts have been widely acknowledged, the likelihood of their most extreme manifestation remains poorly understood.”

Reviewer:

“Is the record-shattering drought threshold of 5% expressed in absolute % or relative %, i.e. calculated with respect to total area of a breadbasket? or area under previous record drought?”

Response:

We thank the reviewer for highlighting this potential ambiguity. The 5% threshold is defined as an absolute percentage exceedance of the previous record value within the same ensemble member, and is calculated with respect to the total breadbasket area rather than with respect to the area affected by the previous record drought. For example, if the previous record drought area in an ensemble member was 10%, then a record-shattering event must reach at least 15%. The drought area itself is defined as the area-weighted percentage of land under drought, calculated either relative to the total land area of each breadbasket or relative to the global maize area, as shown in the figures. The 5% margin therefore represents a substantial, fixed absolute exceedance of the previous record within each model realization.

Given the referee’s comment, to properly clarify this point for the readers, we have revised the corresponding definition in the Methods section. **“A spatially compounding, global record-shattering drought event is defined as a global annual drought area that exceeds the previous maximum within a given ensemble member by an area equal to or larger than 5% of the total breadbasket area (Figure 1a), representing a substantial absolute exceedance of the previous record.”**

Reviewer:

“why some analyses are carried out for the period after 2026? and some for the period after 2000”

Response:

We appreciate the opportunity to clarify the rationale for using different starting years in the analyses. The two starting years serve different analytical purposes. The analyses beginning in 2026 (Figure 4a,b) are designed to quantify the future probability of experiencing at least one global record-shattering drought, which allows us to present both near-term (2026–2055) and long-term (2026–2099) probabilities of such events.

In contrast, analyses beginning in 2000 (Figure 4c-e) serve to generally characterise record-shattering drought events. While we acknowledge that the choice of starting year is somewhat subjective, the inspected characteristic of global record-shattering events (that is, whether they coincide with underlying moderate extremes or record-shattering regional droughts) is not expected to depend on the time of occurrence. Therefore, analysing record-shattering events across the entire 21st century allows us to consider a larger set of events and increase the sample size (compared to an analysis starting in 2026), enabling a more robust characterisation of the record-shattering droughts. We clarified this aspect in the Methods, where we state: “we recomputed the probability that at least one global record-shattering drought occurs during the **future** time periods (2026-2055) and (2026-2099)” and “In Figure 4c-e and Extended Data Figure 4c-e, we inspect whether global record-shattering droughts occurring between 2000 and 2099, **which increases the sample size relative to analyses starting in 2026 and enables a more robust assessment.**”

Reviewer:

“what is the motivation for using +/- 15year window in the analyses of global vs. regional drought? Why not just +15 years? A regional drought occurring after a global one has likely a stronger impact than the other way around, no?”

Response:

Here it is relevant to clarify the rationale for our choice. In the paper, Figure 4c shows that most global record-shattering droughts do not coincide with regional record-shattering events in the same year. Figure 4d demonstrates that global record-shattering droughts instead often emerge from moderately extreme regional droughts. To ensure that this behaviour is not simply due to the potential constraint that there are no regional record-shattering events at all around the year of occurrence of the global record-shattering event, Figure 4e (which is the one the referee refers to and is based on ± 15 -year window) serves to demonstrate that regional record-shattering events still tend to occur, but they do not coincide with global record-shattering events.

The above should clarify that the choice of the window is not related to how regional record-shattering events lead to impacts in relation to global events. In practice, we analyze regional record-shattering events within a ± 15 -year window of each global event. Such a symmetric window allows us to capture both cases in which a regional extreme precedes the global event and cases in which it follows it, both of which matter for informing on whether regional record-shattering droughts occur around the time of a global record-shattering drought.

We address this point by adding the following clarification directly to the Methods: “The case (3) serves to ensure that the finding that global record-shattering droughts often emerge from moderately extreme regional droughts is not simply due to the absence of regional record-shattering events around the year of occurrence of each global event. To this end, we quantified the fraction of global record-shattering droughts for which a regional record-shattering drought occurs within 15 years before or after the global event. Since the global events span

from 2000 to 2099, this window-based analysis is restricted to the period 2000–2085 to ensure that the full ± 15 -year window lies within the study period.”

Reviewer:

“Global drought is calculated as the area-weighted percentage of regional droughts. Shouldn't it be more sound to calculate it as a production-weighted percentage?”

Response:

We understand the referee's interesting point. We note that by focusing on the six major maize-producing regions rather than on the total global land area, our approach already provides a tailored focus on relevant regions. In practice, our approach effectively applies a binary weighting that separates maize-producing regions from non-producing regions, which conceptually connects to the production-weighted framing suggested by the reviewer. We also note that, for a projection-focused study such as ours, implementing a fully production-weighted metric would be highly challenging, as it would require future production estimates that are themselves highly uncertain and strongly influenced by climatic variability.

References

- Bevacqua, E., Suarez-Gutierrez, L., Jézéquel, A., Lehner, F., Vrac, M., Yiou, P., & Zscheischler, J. (2023). Advancing research on compound weather and climate events via large ensemble model simulations. *Nature Communications*, *14*(1), 2145. <https://doi.org/10.1038/s41467-023-37847-5>
- Bevacqua E, Fischer E, Sillmann J & Zscheischler J. Moderate global warming does not rule out extreme global climate outcomes. *Nature* (in press, 2026).
- Fischer, E. M., Beyerle, U., Bloin-Wibe, L., Gessner, C., Humphrey, V., Lehner, F., Pendergrass, A. G., Sippel, S., Zeder, J., & Knutti, R. (2023). Storylines for unprecedented heatwaves based on ensemble boosting. *Nature Communications*, *14*(1), 4643. <https://doi.org/10.1038/s41467-023-40112-4>
- Fischer, E. M., Sippel, S., & Knutti, R. (2021). Increasing probability of record-shattering climate extremes. *Nature Climate Change*, *11*(8), 689-695. <https://doi.org/10.1038/s41558-021-01092-9>
- Liu, J., Chen, J., Yin, J., & King, A. D. (2025). Time of Emergence of Record-Shattering Compound Heatwave-Extreme Precipitation Events and Their Socio-Economic Exposures. *Geophysical Research Letters*, *52*(16), e2025GL116884. <https://doi.org/https://doi.org/10.1029/2025GL116884>
- Anderson, W. B., Seager, R., Baethgen, W., Cane, M., & You, L. (2019). Synchronous crop failures and climate-forced production variability. *Science Advances*, *5*(7), eaaw1976. <https://doi.org/doi:10.1126/sciadv.aaw1976>
- Bevacqua, E., Suarez-Gutierrez, L., Jézéquel, A., Lehner, F., Vrac, M., Yiou, P., & Zscheischler, J. (2023). Advancing research on compound weather and climate events via large ensemble model simulations. *Nature Communications*, *14*(1), 2145. <https://doi.org/10.1038/s41467-023-37847-5>
- de Vries, I., Sippel, S., Zeder, J., Fischer, E., & Knutti, R. (2024). Increasing extreme precipitation variability plays a key role in future record-shattering event probability. *Communications Earth & Environment*, *5*(1), 482. <https://doi.org/10.1038/s43247-024-01622-1>

- Fischer, E. M., Beyerle, U., Bloin-Wibe, L., Gessner, C., Humphrey, V., Lehner, F., Pendergrass, A. G., Sippel, S., Zeder, J., & Knutti, R. (2023). Storylines for unprecedented heatwaves based on ensemble boosting. *Nature Communications*, *14*(1), 4643. <https://doi.org/10.1038/s41467-023-40112-4>
- Fischer, E. M., Sippel, S., & Knutti, R. (2021). Increasing probability of record-shattering climate extremes. *Nature Climate Change*, *11*(8), 689-695. <https://doi.org/10.1038/s41558-021-01092-9>
- Li, B., Liu, K., Wang, M., Yang, Y., He, M., Wang, Y., Li, C., & Wang, D. (2026). Understanding the Dynamics of Record-Shattering Compound Drought-Heatwave Events and Their Impacts on Ecosystems. *Earth's Future*, *14*(1), e2024EF005714. <https://doi.org/https://doi.org/10.1029/2024EF005714>
- Bevacqua, E., Rakovec, O., Schumacher, D. L., Kumar, R., Thober, S., Samaniego, L., Seneviratne, S. I., & Zscheischler, J. (2024). Direct and lagged climate change effects intensified the 2022 European drought. *Nature Geoscience*, *17*(11), 1100-1107. <https://doi.org/10.1038/s41561-024-01559-2>
- Kiros, F. G. (1991). Economic Consequences of Drought, Crop Failure and Famine in Ethiopia, 1973-1986. *Ambio*, *20*(5), 183-185. <http://www.jstor.org/stable/4313818>
- Liu, X., Pan, Y., Zhu, X., Yang, T., Bai, J., & Sun, Z. (2018). Drought evolution and its impact on the crop yield in the North China Plain. *Journal of Hydrology*, *564*, 984-996. <https://doi.org/https://doi.org/10.1016/j.jhydrol.2018.07.077>
- Li, Y., Guan, K., Schnitkey, G. D., DeLucia, E., & Peng, B. (2019). Excessive rainfall leads to maize yield loss of a comparable magnitude to extreme drought in the United States. *Global Change Biology*, *25*(7), 2325-2337. <https://doi.org/https://doi.org/10.1111/gcb.14628>
- Buras, A., Rammig, A., & Zang, C. S. (2020). Quantifying impacts of the 2018 drought on European ecosystems in comparison to 2003. *Biogeosciences*, *17*(6), 1655-1672. <https://doi.org/10.5194/bg-17-1655-2020>
- Kelder, T., Heinrich, D., Klok, L., Thompson, V., Goulart, H. M. D., Hawkins, E., Slater, L. J., Suarez-Gutierrez, L., Wilby, R. L., Coughlan de Perez, E., Stephens, E. M., Burt, S., van den Hurk, B., de Vries, H., van der Wiel, K., Schipper, E. L. F., Carmona Baéz, A., van Bueren, E., & Fischer, E. M. (2025). How to stop being surprised by unprecedented weather. *Nature Communications*, *16*(1), 2382. <https://doi.org/10.1038/s41467-025-57450-0>
- Thompson, V., Kennedy-Asser, A. T., Vosper, E., Lo, Y. T. E., Huntingford, C., Andrews, O., Collins, M., Hegerl, G. C., & Mitchell, D. (2022). The 2021 western North America heat wave among the most extreme events ever recorded globally. *Science Advances*, *8*(18), eabm6860. <https://doi.org/doi:10.1126/sciadv.abm6860>
- Singh, D., Seager, R., Cook, B. I., Cane, M., Ting, M., Cook, E., & Davis, M. (2018). Climate and the Global Famine of 1876–78. *Journal of Climate*, *31*(23), 9445-9467. <https://doi.org/https://doi.org/10.1175/JCLI-D-18-0159.1>
- Singh, J., Ashfaq, M., Skinner, C. B., Anderson, W. B., Mishra, V., & Singh, D. (2022). Enhanced risk of concurrent regional droughts with increased ENSO variability and warming. *Nature Climate Change*, *12*(2), 163-170. <https://doi.org/10.1038/s41558-021-01276-3>
- Qi, W., Feng, L., Yang, H., & Liu, J. (2022). Increasing Concurrent Drought Probability in Global Main Crop Production Countries. *Geophysical Research Letters*, *49*(6), e2021GL097060. <https://doi.org/https://doi.org/10.1029/2021GL097060>
- Singh, J., Ashfaq, M., Skinner, C. B., Anderson, W. B., & Singh, D. (2021). Amplified risk of spatially compounding droughts during co-occurrences of modes of natural ocean variability. *npj Climate and Atmospheric Science*, *4*(1), 7. <https://doi.org/10.1038/s41612-021-00161-2>

Response to Reviewer2

We thank the referee for the careful and constructive review. The comments and suggestions were valuable and have improved the manuscript. We have revised the paper accordingly. Reviewer comments are shown in *italics*, responses in regular text, and changes to the manuscript in **bold**. In addition, reviewer comments have been numbered to enable a clearer correspondence between individual comments and our responses.

Reviewer:

“The study relies entirely on model simulated trends in drought area. While the study briefly touches on the questions of uncertainty and the interannual variability component, there is no discussion, nor even a mention, of this ability of these GCMs to capture an observed record-breaking or “record-shattering” event. Indeed, the wide range from the GCM ensembles in the historical periods shown in Extended Data Figure 1 for example suggest that some of these GCMs likely do not capture the historical frequency of such events, or (more to the definition the authors use) the historical drought area. Simply performing an analysis of trends in drought area from model data and providing no insight to the validity of the model simulations for the target metrics gives no guidance for the validity of the conclusions made by the authors. There should be some evaluation of how well this ensemble of GCMs replicates observed trends in drought area and the frequency of record breaking droughts. Also, since the six GCMs used may replicate relevant physical processes better in some regions than others, it is quite relevant to see who well each GCM captures the drought area in each of the breadbasket regions.”

Response:

We agree that a comparison against observations would strengthen our results. Evaluating model performance for rare, record-shattering soil moisture drought events is challenging, but we agree that such an effort is very important. First, as the reviewer notes, our drought metric is defined relative to a historical reference period (1850–1899), but no observational soil moisture datasets exist for this period, making a direct evaluation of model performance over the reference baseline impossible. Second, given that we have only one realization of the past (observations), it is not possible to robustly evaluate probabilities of record-shattering or record-breaking events, as these events are very rare. We tackled these challenges as described below. The evaluation shows that models are compatible with observations, as detailed below.

To address the data limitation, for the model evaluation, we employed three widely used observational and reanalysis soil moisture datasets that provide complementary information and span multiple decades:

1. ERA5 surface soil moisture (first layer, ~7 cm depth; reanalysis)
2. GLEAM root-zone soil moisture (~0–10 cm; satellite-driven retrievals and land-surface modeling)

3. MERRA-2 surface soil moisture (~5 cm; reanalysis)

These three datasets differ in methodology (reanalysis vs. satellite-driven), forcing, and model structure, and together offer a representative range of observational uncertainty (Gelaro et al., 2017; Hersbach et al., 2020; Martens et al., 2017). Their soil-layer depths are also broadly consistent with the SMILEs' surface soil moisture variable (~0–10 cm; mrsos).

We first followed the reviewer's indication of assessing trends in drought area (Response Figures 1,2,3,4). Among the available datasets, ERA5 provides the longest continuous record. We therefore first compare the SMILEs under SSP2-4.5 with ERA5 over 1950–2024 (Response Figures 1,2), using 1950–1980 as a reference period. During the reference period, the ERA5 trend lies within the SMILEs ensemble spread across all regions; after 2000, this consistency persists at the global scale and for the majority of regions, with only China and Argentina exhibiting differences. While some models are not compatible with ERA5, and for China and Argentina the trend in the drought area is incompatible with ERA5 in general (Response Figure 2), given the large observational uncertainty (see below), it is key to account for all three observational soil moisture products to make robust statements on possible model discrepancies. We account for such observational uncertainty in the evaluation in the following.

Response Figure 1. Comparison of time series of drought area (%) in ERA5 and SMILEs. a-f, time series of drought area for six regions. **g** is for the global scale. Thin grey lines show individual ensemble members from SMILEs models under the SSP2-4.5, with each ensemble smoothed using a ± 5 year moving window to suppress interannual variability. Observed drought area derived from ERA5 is shown as thick black and smoothed using the same ± 5 year moving window for consistency with model simulations.

Response Figure 2. Regional and global drought area trends in ERA5 and SMILEs under SSP2–4.5. **a–g,** the distribution of linear trends in drought area (%) across ensemble members of individual SMILEs models for each region and globally. The black dot shows the observational trend from ERA5. Error bars represent uncertainty in the observational trend, quantified as the standard deviation of the linear regression slope fitted to annual drought area time series over 1950–2024.

The overlapping temporal coverage for the three observational datasets begins in 1980, we thus used 1980–2009 as a common reference period for the evaluation and computed annual drought area for 1980–2024 for each dataset and for each SMILE model. In such a comprehensive multi-observation evaluation, to provide information on record-shattering (and record-breaking) probabilities, we expanded the assessment to other variables. First, we counted the actual numbers of global record-breaking and record-shattering droughts in models and observations. We consider both types of events as record-breaking events are less rare than record-shattering ones, thus providing a larger sample for comparison between datasets. Record-shattering events are very rare: over the last 15 years (2009–2024), relative to the 1980–2009 reference period, they occur only 0–2 times in the SMILEs and just once in each observational dataset. Despite the scarcity of the events, this comparison shows that models are compatible with observations. For record-breaking events: over the same period, the SMILEs yield 0–4 record-breaking events, while the observational estimates are 3 in ERA5, 6 in GLEAM, and 1 in MERRA-2. These results show that model simulations are compatible with two out of three observational datasets (models within observational range). Furthermore, the differences across observations indicate substantial observational uncertainty (see below). While such counts provide useful information, the rarity of these events makes assessing counts difficult and probability challenging, therefore we move to derive additional information. Specifically we evaluate trends and the interannual variability (standard deviation) of the drought area. In fact, information on the probability of record-shattering events can be derived by considering that the occurrence of such events is determined by the combination of the trends in drought area relative to the interannual variability standard deviation of the drought area (for example, a large trend in drought area would not cause many record-shattering events if the standard deviation of the drought area was relatively very large).

The evaluation results for trends and standard deviation (Response Figure 3, 4, 5) reveal that each SMILE model agrees with at least one observational dataset in every region, as well as at the global scale. The spread across observations (ERA5, GLEAM, and MERRA-2) is sufficiently large that model simulations generally fall within it. Given such large differences across observational data, no single dataset can be taken as the reference truth; hence it is not possible to identify one specific model as clearly biased. This implies that even if model-based trends differ from each other—which is consistent with the fact that drought area (defined based on the preindustrial reference period) differ substantially across models in the middle of the last century and therefore at least some models should be biased, as noted by the referee—no robust information is available from observations to label a model as biased.

In some regions, models are consistent with the most optimistic trends across the observational estimates, which implies that if the most pessimistic observational trend is the most realistic, our projected probabilities may be conservative. Overall, both the models and the observational datasets span a wide range of historical drought area trends, underscoring even more the importance of accounting for model uncertainty in projections and transparently communicating it.

To address this comment, we added a part to the final section of the Discussion stating that “The prospect of a widespread global record-shattering drought across major maize breadbaskets underscores the importance of integrating hazard assessments with agricultural impact assessments. **Such assessments should carefully account for climate model uncertainties, particularly given that each of the considered SMILEs agrees with at least one of three observational and reanalysis soil moisture datasets in terms of drought area variability and trends over the last decades—the two key drivers of record-shattering probabilities (Extended Data Figure 12,13; see Methods)**”.

Furthermore, we added details on the evaluation, including a description of the three observational and reanalysis soil moisture datasets in the Data section, and in a new Methods subsection entitled “**Evaluation against observational and reanalysis soil moisture datasets**”. In such a new section, we wrote: “**In general, the evaluation of the probability of record-shattering events is challenged by the limited temporal coverage of observations and the rarity of the events. For this reason, here, we compared SMILEs simulated drought area variability (standard deviation) and trends which are the two key drivers of record-shattering probabilities, against observations and reanalysis. Furthermore, because drought is defined relative to a preindustrial reference period (1850-1899), direct observation by taking into account the baseline preindustrial period is not possible. We therefore assess the realism of simulated drought area by comparing SMILEs with observational and reanalysis datasets over the overlapping period of data availability, that is, 1980-present. Accordingly, we select 1980-2009 as a reference period to define grid-point droughts and compute annual drought area for 1980-2024 for each dataset and each SMILE model.**

This approach enables a consistent evaluation of the interannual variability and long-term trends in drought area at both global and regional scales (six major breadbasket regions). Linear trends in annual drought area (%) are estimated for each ensemble member and summarised at the model level for comparison with observational estimates. The uncertainty ranges shown for the observational and reanalysis datasets in Extended Data Figure 12 correspond to the standard deviation (σ) of the estimated drought-area trends. Interannual variability is quantified as the standard deviation (σ) of annual drought area over the evaluation period (Extended Data Figure 13).

Overall, the comparison indicates that each SMILE model agrees with at least one observational dataset in every region and globally, with simulations generally falling within the spread of observational uncertainty (Extended data Figure 12,13). These results highlight the importance of carefully considering the uncertainties in our projected

probabilities of record-shattering droughts, even more as a broader sampling of models could further expand the range of projections (Bevacqua et al., 2026).”

Response Figure 3. Same as Response Figure 1, but including three observational datasets for period 1980-2024. Observed drought area time series derived from ERA5, GLEAM, and MERRA-2 are shown as thick black, blue, and green lines, respectively. As in Response Figure 1, the drought area time series are smoothed using a ± 5 year moving window.

Response Figure 4. Same as Response Figure 2, but with observational trends from ERA5 (black), GLEAM (blue), and MERRA-2 (green) shown by symbols for the period 1980–2024.

Response Figure 5. Same as Response Figure 4, but for the standard deviation (SD) of the drought area.

Reviewer:

“Related to the above, [1] there is no justification given for why the six specific GCMs are used. Perhaps this is related to the ability of the GCMs to capture critical processes related to soil moisture and drought, but the reader is left to wonder since there is no justification given. [2] In a similar vein, the authors describe their drought classification as being when the annual surface soil moisture falls below the 1850-1899 local 20th percentile. An interesting choice for how to classify a drought, but why use the 20th percentile? Why not the 10th or 5th percentile? [3] Similarly, why is a “record-shattering” drought when an event exceeds previous drought area values by at least 5%? Why not 2.5% or 10% or another value? Also, why the 10cm soil moisture over the soil moisture from other depths? [We address this last question in point 2 below.] The appropriate justification for each of these should be included in this manuscript. In addition, my questions, among others also present useful opportunities for future work related to the sensitivity of the results. For example, if the threshold of a drought was the local 5th percentile over the 20th percentile, would the results presented here be significantly different? [We address this question in point 2 below.] Such options for future work and analysis could be discussed in the results section.”

Response:

We added numbers to the referee’s comment above so as to refer to each aspect individually. We thank the reviewer for raising these related and important points about model selection, drought definitions, threshold choices, and the sensitivity of the results. In the following, we address each of these points and discuss their implications for the robustness and interpretation of the results. In addition to providing specific information on the analysis in this work, we agree that in line with the reviewer’s view these additions provide useful insights for future works on similar topics.

1. Why we choose the six specific SMILEs models.

Estimating the probability of record-shattering compound droughts requires a large number of model realizations. Within the CMIP6 archive, only six models provide more than ten ensemble members over the period of interest, which we consider the minimum ensemble size needed to study the probability of record-shattering events. These six SMILEs were therefore selected based on ensemble-size availability rather than model performance. Although this constraint limits the number of models, the selected SMILEs span a range of model structures and physical parameterizations, providing a meaningful representation of model spread while meeting the ensemble-size requirements of our analysis.

To clarify this aspect to the reader, we added a sentence in the Methods section stating that **“Previous studies show that robust probability estimates for compound events and record-shattering events require a sufficiently large number of realizations (Bevacqua et al., 2023; Fischer et al., 2023; Fischer et al., 2021); accordingly, we selected models with at least 10 ensemble members, which we consider sufficient for reliable probability estimation”**.

2. Why we use surface soil moisture. Why the threshold for drought is defined as the 20th percentile of surface soil moisture.

We focus on surface soil moisture because it directly governs plant water stress and land–atmosphere fluxes (Fu et al., 2022), and is therefore highly relevant for identifying drought impacts in a changing climate. Surface soil moisture also provides a physically grounded drought indicator that is directly available from all SMILEs, making it well-suited for a model-based analysis of rare and compound drought extremes.

To clarify our choice of surface soil moisture, we added the following text to the Methods: **“Surface soil moisture provides a physically grounded indicator of agricultural drought, as it directly governs plant water stress and land–atmosphere fluxes and is therefore closely linked to drought impacts in a changing climate (Fu et al., 2022; Korres et al., 2013).”**

The 20th percentile threshold is widely used to define moderate drought in hydrological and agricultural applications and is consistent with the D1 (“moderate drought”) category of the U.S. Drought Monitor (Buitink et al., 2021; Monitor, 2023; Osman et al., 2024; Řehoř et al., 2025). This threshold provides a balance between representing spatially extensive drought conditions and retaining sufficient event frequency to ensure a robust statistical analysis.

To clarify our drought definition, we added the following sentence to the Methods: **“This percentile-based definition corresponds to moderate drought conditions as commonly defined in drought classification frameworks and provides a consistent threshold for identifying spatially extensive droughts while retaining a sufficiently large sample size for robust statistical analyses (Osman et al., 2024; Řehoř et al., 2025)”**.

In addition, to assess the sensitivity of our results to the drought definition, we repeated the analysis using a more restrictive 5th percentile threshold, representative of extreme drought conditions. Under the SSP2-4.5 scenario a comparison of Figure 1c in the main text (20th percentile) with Response Figure 6a (5th percentile) shows that, as expected for a more restrictive definition of drought, the absolute drought area is smaller and interannual variability is larger when using the 5th percentile threshold. The time behaviour of the probability of record-shattering events is similar under both thresholds (Figure 1c in the main text and Response Figure 6b); this aligns with the fact that record-shattering events are identified based on relative exceedances of previous maxima within each definition, rather than on the absolute magnitude of drought area.

Using a more restrictive drought definition (5th percentile; shown in Response Figure 7c) changes the partitioning in Figure 4c of the manuscript: global record-shattering droughts more frequently coincide with a single region undergoing a regional record-shattering drought, compared to the 20th percentile definition (Response Figure 7c). However, importantly, multi-region synchronized regional record-shattering remains uncommon, thus the main conclusion remains unchanged: global record-shattering droughts generally emerge from

widespread moderately extreme drought conditions across multiple regions rather than from synchronized regional record-shattering events (Response Figure 7c,d; the partitioning in Response Figure 7d remains comparable to that of Figure 4d). Furthermore, the dominant regional contributors to the probability of experiencing at least one global record-shattering drought remain unchanged (Response Figure 7a,b), highlighting the robustness of the identified regional drivers. Overall, these results (Response Figure 6 and 7) confirm that the main conclusions of the study are robust to the choice of drought definition.

To address this comment, we revised the Results to explicitly assess the sensitivity of our conclusions to the drought severity. We added some description to: “However, results under SSP2-4.5 reveal that approximately 73% of global record-shattering droughts occur without any underlying regional record-shattering event (Figure 4c). **This fraction increases to about 82% under SSP5-8.5 (Extended Data Figure 4c), and decreases to approximately 42% when defining local droughts based on the more restrictive 5th percentile threshold rather than the 20th percentile (Extended Data Figure 5c).** When a regional record-shattering drought is involved, it is typically confined to a single region—mainly Europe, the USA, or Brazil. Consequently, global record-shattering droughts are rarely driven by two or more regions simultaneously under regional record-shattering droughts (2.1% under SSP2-4.5; 1.3% under SSP5-8.5; and 6.3% under SSP2-4.5 with the 5th percentile definition).”

We also rephrase the statement in the final paragraph to: “**These findings are robust against alternative drought definitions, different thresholds for moderate regional droughts, and different emission pathways. Specifically, analyses using a more restrictive 5th percentile drought threshold, varying the definition of moderate regional drought, and considering the SSP5–8.5 scenario yield qualitatively similar behaviour (Extended Data Figures 4, 5 and 6), underscoring that, regardless of drought severity or future emissions, global record-shattering droughts are generally expected to emerge from the aggregation of moderately extreme regional droughts rather than from perfectly synchronized regional record-shattering events.**”.

Response Figure 6. The evolution of the regional and global drought areas for the major maize-producing regions and the annual probability of global record-shattering drought under SSP2-4.5 with a 5th percentile threshold for drought. a. Stacked multi-model mean of the annual drought area for the six regions and globally, defined as the percentage of the global maize area under drought. The error bars for the global drought area indicate the inter-model range. **b.** Probability during 1950-2085 based on the original data (non-detrended), where the solid red line shows the multi-model mean and red shading shows the inter-model range. Other lines show the contribution to the probabilities from changes in surface soil moisture's standard deviation (SD, yellow), mean (blue), and both mean and standard deviation (grey).

SSP2-4.5

Response Figure 7. Contributions of regional surface soil moisture trends and timing of regional droughts relative to global events under SSP2-4.5 with a 5th percentile threshold for drought. a, Multi-model mean probability of the occurrence of at least one global record-shattering drought during 2026-2055 (red bar) and the same probability under stationary surface soil moisture in all crop regions (gray bar). The other bars represent the contribution of soil-moisture trends in each crop region. Error bars represent inter-model range. **b,** As panel a, but for the probability during 2026-2099. **c,** Multi-model mean of the proportion of global record-shattering droughts during 2000-2099 that coincide with no regions, only one region ('Rest only' means the sum of the remaining cases of only one region), or multiple regions simultaneously under regional record-shattering droughts — categories with 0% proportions are not shown. **d,** As panel c, but for cases where global record-shattering droughts coincide with regional moderately extreme droughts. **e,** Multi-model mean of the proportion of global record-shattering droughts accompanied by a record-shattering drought in a given region (x-axis) within 15 years before or after the global event year; error bars represent inter-model range.

3. Why we define the record-shattering threshold as 5%.

We thank the reviewer for raising this relevant point regarding the choice of the record-shattering margin. To assess the sensitivity of our results to this choice, we performed additional analyses using thresholds around 5%. When the threshold is increased to 7.5%, the probability of global record-shattering droughts becomes extremely small across all models (typically below 0.5% over 1950–2100; see Response Figure 8). This reflects the fact that a 7.5 % exceedance lies well beyond the range of interannual variability in drought area simulated by the models, hence a 7.5% does not provide a useful metric for distinguishing exceptionally large drought area

anomalies. Note that to ensure that our probability estimates are statistically robust for such rare events, we also applied a bootstrap approach (Methods), which confirms that probabilities remain extremely small for a 7.5% threshold.

In contrast, setting the margin to 2.5% produces the opposite behavior (Response Figure 9): the estimated probabilities rise substantially in all models. A 2.5% exceedance falls well within the typical range of interannual fluctuations and leads to results that tend to be more similar to those obtained for ordinary record-breaking events. Taken together, we chose a 5% margin because it is large enough to capture record-shattering events, while still clearly separating them from more modest record-breaking ones.

Response Figure 8. Annual probability of global record-shattering droughts with a threshold of 7.5%. a-f, Probability under SSP2-4.5 based on different models (stated in the title of the panels). The red line shows the probability based on the original (non-detrended) data (consistent with other record-shattering probabilities in the paper, based on pooled ensemble members of each model). For each model, the shading shows the centered 95% confidence intervals derived via bootstrap (1000 resamples of the ensemble members of the model) to quantify the uncertainty from internal variability.

Response Figure 9. Same as Response Figure 8 but for a 2.5% threshold.

To further assess robustness, we repeated the regional contribution analysis shown in paper Figure 4a, b using the alternative 7.5% and 2.5% margins (Response Figure 10). Despite the substantial differences in event probabilities under these two settings (Response Figures 8 and 9), the multi-model probability patterns remain consistent: in all cases, soil moisture trends in Brazil, Europe, and the USA emerge as the dominant drivers of the global record-shattering drought. These results confirm that the key attribution findings presented in the paper are stable and do not depend on the exact threshold used to define a record-shattering event.

To address this comment, we added the following sentence to the Methods section: “**Sensitivity analyses illustrate that the choice of exceedance margin affects the characterization of record-shattering droughts: a higher margin (7.5%) suppresses nearly all events across models (Extended Data Fig. 10a–f), whereas a lower margin (2.5%) yields probabilities comparable to ordinary record-breaking droughts (Extended Data Fig. 10g–l). However, the probability patterns remain consistent: in all cases, soil moisture trends in Brazil, Europe, and the USA emerge as the dominant drivers of the global record-shattering drought (Extended Data Figure 11). The chosen 5% margin balances these cases, allowing**

for capturing record-shattering events while providing a clear separation from record-breaking ones.”

Response Figure 10. **a**, Multi-model mean probability of the occurrence of at least one global record-shattering drought with a threshold of 7.5% during 2026-2055 (red bar) and the same probability under stationary surface soil moisture in all crop regions (gray bar). The other bars represent the contribution of soil-moisture trends in each crop region. Error bars represent inter-model range. **b**, As panel a, but for the probability during 2026-2099. **c,d**, As **a,b** but for the threshold of 2.5%.

Reviewer:

“The key conclusions depend on how the authors define the probability of a record-shattering event. First, the authors define a record-shattering event as being related to drought area, rather than the drought intensity. This is interesting, and I do wonder why this choice to focus on

drought area for this definition without regard to intensity. Under this definition, a record-shattering drought could be one that covers a significantly large area, but where the classification for drought in all of the region effected is barely met. Is it really record-shattering if the area effected is larger, but the intensity does not change (or changes little)? There was no explanation given for the focus on area over intensity, or better still, area alongside intensity. Please add an explanation for that to the methods.”

Response:

We thank the referee for this comment, which allowed us clarify some interesting aspects in the revised paper. In this study, the term “record-shattering” is defined explicitly with respect to drought spatial extent, rather than local drought intensity. Under this definition, a drought that affects a fraction of the land area that is 5% larger than in previous events would indeed be classified as record-shattering. This reflects our focus on spatially extensive and synchronized drought conditions, which constitute a distinct class of climate risk by limiting spatial buffering and amplifying large-scale societal and ecological impacts.

Note, however, that drought spatial extent and drought intensity are usually not independent. As shown in new analyses (Response Figure 11), the global drought area is strongly negatively correlated with area-averaged soil moisture anomalies across both historical (1850–1899) and future (2000–2049 and 2050–2099) periods, indicating that increasingly large drought extents generally coincide with enhanced mean drought intensity. Inspired by the referee’s comment, we believe that it is very interesting and relevant to highlight this aspect in the paper.

To further clarify the relationship between drought extent and intensity, we added a sentence to the Discussion stating that “**Furthermore, drought spatial extent and drought intensity are tightly coupled (Extended Data Figure 9), meaning that our considered record-shattering spatially extended droughts are expected to coincide with substantial negative anomalies in soil moisture**”.

We also added a corresponding description in the Methods (in the subsection “Global and regional record-shattering droughts, and their probability”): “**In our analysis, we focus on the drought area. In Extended Data Figure 9, to characterize drought intensity associated with spatially compounding droughts, we define the global mean drought intensity as the area-weighted mean of standardized surface soil moisture anomalies across all grid cells classified as under drought in a given year. Surface soil moisture anomalies were computed relative to the preindustrial reference period (1850–1899) and standardized by dividing by the corresponding standard deviation (σ) at each grid cell.**”.

Response Figure 11. Association between drought spatial extent and mean drought intensity. a-f, Scatter plot of annual global drought area versus the area-weighted mean of standardized surface soil moisture anomalies across all grid cells under drought, for individual models. Each point represents the annual value from one ensemble member. Colors denote three time periods (1850–1899, 2000–2049, and 2050–2099).

Reviewer:

“Second, the definition of the annual probability of a record-shattering event is perhaps incorrect (perhaps in this case because the awkward phrasing of the definition has left me unclear). The authors state that the probability each year, with the 31-year moving window, is “The number of record-shattering events is divided by the total number of annual drought area values across all ensemble members (that is, 31 times the number of the considered ensemble members), yielding a time-varying probability of record-shattering events.” This sentence is quite confusing, but the interpretation I had reading this is that the probability is defined as the number of record-shattering events divided by the number of drought events. If this is correct, then the actual value defined is the probability of a given drought event being a record-shattering event, rather than a probability of a record-shattering drought occurring at all. This definition needs to be clarified for readers, and analysis of the probability should reflect this definition.”

Response:

We agree that the original phrasing can be misinterpreted. We have therefore improved the definition by revising the text to clarify that the probability represents the fraction of years under record-shattering droughts relative to all years (and not relative only to years under drought).

We now replace “the total number of annual drought area values” with “years” to avoid misunderstandings: “To estimate the annual probability of record-shattering events (Figure 3 and Extended Data Figure 7), a 31-year moving window (± 15 years) is used. Within each window, the number of record-shattering events is divided by the total number of **years** across all ensemble members (that is, 31 times the number of the considered ensemble members), yielding a time-varying probability of record-shattering events.”

Reviewer:

“Lines 55-57: “This is shown by a comparison of the actual changes...” – The figures are not showing the actual observed changes, but the modeled changes from one random model. Please adjust the text, but also describe which model is used for Figure 1a.”

Response:

We have revised the text to clarify that the comparison refers to “model-simulated” changes in drought area rather than actual observed changes. We also now explicitly state that Figure 1a shows results from the “MIROC6” model.

Reviewer:

“Lines 92-94: “Retaining surface soil moisture...increase in probability.” – Why detrend some but not all?”

Response:

Our analysis includes a detrend-all experiment that serves as the baseline (see Methods). Retaining the trend in one region at a time allows us to assess that region’s contribution to the global probability relative to this baseline.

References:

- Bevacqua, E., Suarez-Gutierrez, L., Jézéquel, A., Lehner, F., Vrac, M., Yiou, P., & Zscheischler, J. (2023). Advancing research on compound weather and climate events via large ensemble model simulations. *Nature Communications*, 14(1), 2145. <https://doi.org/10.1038/s41467-023-37847-5>
- Buitink, J., van Hateren, T. C., & Teuling, A. J. (2021). Hydrological System Complexity Induces a Drought Frequency Paradox [Original Research]. *Frontiers in Water*, Volume 3 - 2021. <https://doi.org/10.3389/frwa.2021.640976>
- Fischer, E. M., Beyerle, U., Bloin-Wibe, L., Gessner, C., Humphrey, V., Lehner, F., Pendergrass, A. G., Sippel, S., Zeder, J., & Knutti, R. (2023). Storylines for unprecedented heatwaves based on ensemble boosting. *Nature Communications*, 14(1), 4643. <https://doi.org/10.1038/s41467-023-40112-4>
- Fischer, E. M., Sippel, S., & Knutti, R. (2021). Increasing probability of record-shattering climate extremes. *Nature Climate Change*, 11(8), 689-695. <https://doi.org/10.1038/s41558-021-01092-9>

- Fu, Z., Ciais, P., Feldman, A. F., Gentine, P., Makowski, D., Prentice, I. C., Stoy, P. C., Bastos, A., & Wigneron, J.-P. (2022). Critical soil moisture thresholds of plant water stress in terrestrial ecosystems. *Science Advances*, *8*(44), eabq7827. <https://doi.org/doi:10.1126/sciadv.abq7827>
- Korres, W., Reichenau, T. G., & Schneider, K. (2013). Patterns and scaling properties of surface soil moisture in an agricultural landscape: An ecohydrological modeling study. *Journal of Hydrology*, *498*, 89-102. <https://doi.org/https://doi.org/10.1016/j.jhydrol.2013.05.050>
- Gelaro, R., McCarty, W., Suárez, M. J., Todling, R., Molod, A., Takacs, L., Randles, C. A., Darmenov, A., Bosilovich, M. G., Reichle, R., Wargan, K., Coy, L., Cullather, R., Draper, C., Akella, S., Buchard, V., Conaty, A., da Silva, A. M., Gu, W., . . . Zhao, B. (2017). The Modern-Era Retrospective Analysis for Research and Applications, Version 2 (MERRA-2). *Journal of climate*, *30*(14), 5419-5454. <https://doi.org/https://doi.org/10.1175/JCLI-D-16-0758.1>
- Hersbach, H., Bell, B., Berrisford, P., Hirahara, S., Horányi, A., Muñoz-Sabater, J., Nicolas, J., Peubey, C., Radu, R., Schepers, D., Simmons, A., Soci, C., Abdalla, S., Abellan, X., Balsamo, G., Bechtold, P., Biavati, G., Bidlot, J., Bonavita, M., . . . Thépaut, J.-N. (2020). The ERA5 global reanalysis. *Quarterly journal of the royal meteorological society*, *146*(730), 1999-2049. <https://doi.org/https://doi.org/10.1002/qj.3803>
- Martens, B., Miralles, D. G., Lievens, H., van der Schalie, R., de Jeu, R. A. M., Fernández-Prieto, D., Beck, H. E., Dorigo, W. A., & Verhoest, N. E. C. (2017). GLEAM v3: satellite-based land evaporation and root-zone soil moisture. *Geosci. Model Dev.*, *10*(5), 1903-1925. <https://doi.org/10.5194/gmd-10-1903-2017>
- Monitor, U. D. (2023). US drought monitor. Retrieved from.
- Osman, M., Zaitchik, B., Otkin, J., & Anderson, M. (2024). A global flash drought inventory based on soil moisture volatility. *Scientific Data*, *11*(1), 965. <https://doi.org/10.1038/s41597-024-03809-9>
- Řehoř, J., Brázdil, R., Rakovec, O., Hanel, M., Fischer, M., Kumar, R., Balek, J., Poděbradská, M., Moravec, V., Samaniego, L., Markonis, Y., & Trnka, M. (2025). Global catalog of soil moisture droughts over the past four decades. *Hydrol. Earth Syst. Sci.*, *29*(14), 3341-3358. <https://doi.org/10.5194/hess-29-3341-2025>